# Chronic kidney disease and the outcomes of fibrinolysis for ST-segment elevation myocardial infarction: A real-world study

Wuxiang Xie[1], Anushka Patel[2], Eric Boersma[3], Lin Feng[1,4], Min Li[5], Runlin Gao[6], Yangfeng Wu[1,4,7]*

**1** Peking University Clinical Research Institute, Peking University First Hospital, Beijing, China, **2** The George Institute for Global Health, University of New South Wales, Sydney, Australia, **3** Department of Cardiology, Erasmus Medical College, Rotterdam, The Netherlands, **4** The George Institute for Global Health at Peking University Health Science Center, Beijing, China, **5** Clinical Epidemiology and EBM Center, Beijing Friendship Hospital, Capital Medical University, Beijing, China, **6** Department of Cardiology, Fuwai Hospital, Chinese Academy of Medical Sciences and Peking Union Medical College, Beijing, China, **7** Department of Epidemiology and Biostatistics, School of Public Health, Peking University Health Science Center, Beijing, China

* wuyf@bjmu.edu.cn

##  OPEN ACCESS

**Data Availability Statement:** Data cannot be shared publicly because the original data involved sensitive information of patients. Data are available on request from the corresponding author (contact

## Abstract

### Background

In low-resource regions, fibrinolytic therapy is often the only option for ST-elevation myocardial infarction (STEMI) patients as primary percutaneous coronary intervention (PCI) is often not available and patients are hardly transferred to a medical center with PCI capacity within the first 120 minutes. Chronic kidney disease (CKD) is one of the most frequently encountered complications of STEMI. However, the evidence for the efficacy of fibrinolytic therapy in STEMI patients with CKD is still limited. The aim of this study is to test whether CKD modifies the association between fibrinolytic therapy and short-term major adverse cardiovascular events (MACEs) among patients with STEMI.

### Methods and findings

This is a real-world study analyzing the data from 9508 STEMI patients (mean age: 64.0 ±12.4 years; male: 70.1%) in the third phase of Clinical Pathways in Acute Coronary Syndromes program (CPACS-3), which is a large study of the management of acute coronary syndromes (ACS) in 101 county hospitals without PCI capacity in China. CKD was defined as an estimated glomerular filtration rate of less than 60 mL/min per 1·73 m$^2$ at the admission. The primary outcome is short-term MACEs, including all-cause death, recurrent myocardial infarction, or nonfatal stroke. Patients were recruited consecutively between October 2011 and November 2014. Out of them, 1282 patients (13.5%) were classified as having CKD. Compared with non-CKD patients, CKD patients were less likely to receive fibrinolytic therapy than non-CKD patients (26.4% vs. 38.9%, *P*<0.001), more likely to experience a failed fibrinolytic therapy (32.8% vs. 16.9%), and had a higher risk of short-term MACEs (19.7% vs. 5.6%). After full adjustment, use of fibrinolytic therapy was associated with a

via wuyf@bjmu.edu.cn) or the Peking University IRB (contact via llwyh@bjmu.edu.cn).

**Funding:** Source of funding used to support the research and creation of the article is from Sanofi, China, through an unrestricted research grant. The George Institute for Global Health at PUHSC sponsored the study and owns the data. However, the authors are solely responsible for the design and conduct of this study, all study analyses, the drafting and editing of the manuscript, and its final contents. The funding source had no role in the design and conduct of the study; collection, management, analysis, and interpretation of the data; preparation, review, or approval of the manuscript; and decision to submit the manuscript for publication.

**Competing interests:** The authors declare no competing interests. There are no other relevant declarations relating to employment, consultancy, patents, products in development or marketed products etc. to be made by Sanofi, China. This does not alter our adherence to PLOS ONE policies on sharing data and materials.

significantly lower risk of short-term MACEs in non-CKD patients (relative risk [RR] = 0.87, 95% confidence interval [CI]: 0.76–0.99), but not in CKD patients (*P* for interaction = 0.026). Further analysis stratified by the success of fibrinolysis showed that compared with patients who did not receive fibrinolytic therapy, patients with successful fibrinolysis had a lower risk of short-term MACEs that was similar between patients with (RR = 0.71, 95% CI: 0.55–0.82) and without CKD (RR = 0.67, 95% CI: 0.55–0.92), while patients with unsuccessful fibrinolysis had a similarly higher risk in CKD patients (RR = 1.25, 95% CI: 1.09–1.43) and non-CKD patients (RR = 1.30, 95% CI: 1.13–1.50).

## Conclusions

CKD reduced the likelihood of successful fibrinolysis and increased the risk of short-term MACEs in patients with STEMI. Attention should be paid to how to improve the success rate of fibrinolytic therapy for STEMI patients with CKD.

## Trial registration

The CPACS-3 study was registered on www.clinicaltrials.gov (NCT01398228).

## Introduction

Myocardial infarction (MI) remains one of the leading causes of death worldwide [1], and ST-elevation myocardial infarction (STEMI) is associated with the greatest case-fatality [2]. Incidence rates of STEMI have declined in Europe and the US [2, 3], but significantly increased in China during the past decade [4]. The latest international and Chinese guidelines recommend that primary percutaneous coronary intervention (PCI) should be the preferred reperfusion therapy for STEMI patients within the first 12 hours of symptom onset, and the time to primary PCI from STEMI diagnosis is recommended to be ≤120 minutes [2, 3, 5]. However, if primary PCI cannot be performed within this timeframe or where access to such treatment is not available, fibrinolytic therapy should be administered to STEMI patients without any contraindications [2, 3, 5]. Although PCI is the standard of care for STEMI in economically developed areas, application of fibrinolysis to the large population of STEMI patients in rural areas where STEMI patients cannot receive timely primary PCI is still very important.

One of the most frequently encountered co-morbidities of coronary disease is kidney disease including chronic kidney disease (CKD) and acute kidney injury (AKI), which is estimated to be present in about one-third patients with STEMI in the US [6, 7]. As CKD is associated with higher risks of bleeding and death, patients with kidney disease are frequently excluded from major randomized trials of cardiovascular disease [8, 9]. Thus, evidence for the efficacy and safety of fibrinolytic therapy in STEMI patients with CKD is still limited [10].

The third phase of Clinical Pathways in Acute Coronary Syndromes program (CPACS-3) is a large study of the management of suspected acute coronary syndromes (ACS) in non-PCI-capable hospitals in China. Data from this study provide an opportunity to test whether CKD modifies the associations between fibrinolysis and short-term outcomes after STEMI.

## Methods

### Study population

CPACS-3 was a stepped-wedge cluster randomized trial among resource-constrained hospitals in China [11]. The details of the rationale and design have been published previously [12]. The

objective of CPACS-3 was to determine whether the routine use of a clinical pathway-based, multifaceted quality of care initiatives (QCI) would lead to a measurable reduction in the number of short-term major adverse cardiovascular events (MACEs) in patients with ACS [12]. From eligible hospitals that agreed to participate, ACS patients were consecutively enrolled in five 6-months cycles. No intervention was applied in the first cycle, after which hospitals were randomly allocated to commence the intervention in one of the 4 remaining cycles. The intervention was applied at the level of hospital, with outcomes measured at the patient level. All patients over 18 years old and with a final diagnosis of ACS at discharge or death were recruited consecutively within each 6 month cycle. We excluded patients who were dead on or within 10 minutes of hospital arrival. Between October 2011 and November 2014, a total of 29 346 patients with ACS were recruited from 101 county hospitals without the capacity to perform onsite PCI, of whom 10 294 patients were diagnosed with STEMI. In these observational analyses, 786 patients (7.6%) were excluded because of missing admission data on serum creatinine concentration. The remaining 9508 STEMI patients were included.

The Peking University IRB reviewed and approved the CPACS-3 trial and all participating patients provided written informed consent. The CPACS-3 study was registered on www.clinicaltrials.gov (NCT01398228).

## Data collection

A dedicated web-based data management system was used to collect data by a trained hospital staff member who was not involved in of the management of patients. Data for each patient were collected from the medical record and from survivors prior to hospital discharge, including socio-demographic information, symptoms and signs relating to the presenting STEMI, medical history, electrocardiographic, biomarkers, treatments administered prior to admission, during hospitalization and at death or hospital discharge, final diagnosis and discharge status, major clinical events, personal insurance status, and the total cost of hospitalization. Data quality was maintained through in person and on-line study monitoring activities.

## Definitions

CKD was defined as an estimated glomerular filtration rate (eGFR) of less than 60 mL/min per 1·73 $m^2$ at the admission, with eGFR calculated using a modified Modification of Diet in Renal Disease (MDRD) equation for Chinese: eGFR = $175 \times Scr^{-1.234} \times age^{-0.179}$ [if female, $\times$ 0.79], where Scr is serum creatinine concentration (in mg/dL) and age in years [13]. Diabetes was defined by a reported medical history of diabetes, or current use of glucose lowering therapy at the time of enrollment. Hypertension was defined by a reported medical history of hypertension, systolic blood pressure (SBP) $\geq$140mmHg or diastolic blood pressure $\geq$90mmHg on hospital presentation. Delay to admission was defined as onset-to-door time >2 hours, and delay to fibrinolytic therapy was defined as onset-to-needle time >12 hours [2, 3, 5]. Successful fibrinolysis was defined as the presence of any two of the following four criteria, excluding "c plus d": a) reduction of elevated ST-segment $\geq$50% by ECG achieved within 60 to 90 minutes of receiving fibrinolytic therapy; b) time to peak cTn concentration $\leq$12 hours from symptom onset and/or time to CK-MB concentration $\leq$14 hours from symptom onset; c) significant relief of chest pain within 2 hours of fibrinolytic therapy; d) presence of reperfusion arrhythmia within 2 to 3 hours of fibrinolytic therapy, including accelerated idioventricular rhythm, sudden improvement or disappearance of atrioventricular block or bundle branch block, and transient sinus bradycardia or sino-auricular block with or without hypotension among patients with inferior wall MI [5, 14]. If thrombolysis was not successful according to above criteria, then it was considered as a failed fibrinolysis.

## Study endpoints

The primary outcome of the study was short-term MACEs, including all-cause mortality, recurrent myocardial infarction, or nonfatal stroke. We chose all-cause mortality rather than cardiac death because our definition of all-cause mortality not only included patients who died in hospital, but also those who were discharged against medical advice and died within 1 week, and those who transferred to upper level hospitals but died within 24 hours, as Chinese people would prefer to die at home after relinquishing curative treatments. The secondary outcome was all-cause mortality, recurrent MI, stroke, or severe bleeding. All outcome events were adjudicated by an independent committee according to predefined criteria [12].

## Covariates

Covariates that might confound the association between fibrinolysis and short-term MACEs were selected in our analyses, including age, sex, education, occupation, current smoking, body mass index, history of diabetes mellitus, hypertension, cardiovascular disease, fibrin-specific thrombolytic agent, delay to admission, delay to fibrinolytic therapy, SBP lower than 90 mmHg when presenting at hospital, heart rate higher than 100 beats/m when presenting at hospital, in-hospital use of aspirin, clopidogrel, angiotensin-converting enzyme inhibitors/angiotensin receptor blockers, β-Blockers, calcium channel blockers, and statins. As CPACS-3 was a trial evaluating the effect of QCI on short-term MACEs, we also included QCI intervention and intervention cycle as covariates.

## Statistical analysis

The results are presented as percentages for categorical variables and means ± standard deviations or medians with interquartile ranges for continuous variables. The characteristics of patients with and without CKD were compared using the t-test, Wilcoxon rank test, or chi-square test. Modified Poisson regression using PROC GENMOD was performed to assess the strength of associations (relative risk, RR) between CKD and outcomes, and to identify the predictors of successful fibrinolysis in patients who received fibrinolytic therapy [15]. We adopted modified Poisson regression rather than Logistic regression as the primary outcome was not rare [15]. Generalized estimating equations were used to adjust for clustering within hospitals. To analyze the data of total participants together, we included CKD (yes, no) and two dummy variables for fibrinolytic therapy (yes, no; one for CKD patients and one for non-CKD patients) in the model [16]. To test whether the presence of CKD modified the associations between fibrinolytic therapy and outcomes, we included CKD, fibrinolytic therapy, and their interaction term in another model.

To ensure robustness of our findings and reduce the impact of treatment-selection bias, additional propensity-based subgroup analyses were performed to test the associations between fibrinolytic therapy and outcomes. In these, patients who did and did not receive fibrinolytic therapy were matched (1:1) on all covariates in Table 1 except delay to fibrinolytic therapy and fibrin-specific thrombolytic agent, using the Greedy matching macro [17]. Best matches are those with the highest digit match on propensity score. First, cases are matched to controls on 8 digits of the propensity score. For those that do not match, cases are then matched to controls on 7 digits. The algorithm proceeds sequentially to the lowest digit match on propensity score (1 digit) [17].

To limit the impact of AKI on our main results, we performed another sensitivity analysis excluding patients with high risk of AKI, who had an SBP <90 mmHg and/or a heart rate ≥100 beats/minute.

**Table 1. Baseline characteristics between participants who did and did not receive fibrinolytic therapy, by the presence of reduced kidney function (eGFR <60 mL/min/1.73 m$^2$).**

| | eGFR ≥60 mL/min/1.73 m$^2$ (n = 8226) | | | eGFR <60 mL/min/1.73 m$^2$ (n = 1282) | | |
|---|---|---|---|---|---|---|
| | No fibrinolysis (n = 5025) | Fibrinolysis (n = 3201) | *P* for difference[†] | No fibrinolysis (n = 944) | Fibrinolysis (n = 338) | *P* for difference[†] |
| Age (years) | 65.2±12.4 | 59.3±11.0 | <0.001 | 72.8±10.5 | 66.3±10.4 | <0.001 |
| Men (%) | 3402 (67.7) | 2503 (78.2) | <0.001 | 538 (57.0) | 221 (65.4) | 0.007 |
| eGFR (mL/min/1.73 m$^2$) | 101 (82–125) | 104 (85–130) | <0.001 | 46 (35–54) | 50 (39–55) | <0.001 |
| Hospital stay (days) | 10 (6–14) | 11 (7–14) | <0.001 | 9 (4–13) | 8 (2–13) | 0.007 |
| Delay to admission (%)* | 3324 (66.2) | 1582 (49.4) | <0.001 | 637 (67.5) | 185 (54.7) | <0.001 |
| Delay to fibrinolytic therapy (%)* | / | 296 (9.3) | / | / | 31 (9.2) | / |
| Fibrin-specific thrombolytic agent (%) | / | 1272 (39.7) | / | / | 140 (41.4) | / |
| Education ≥High school (%)* | 472 (9.4) | 460 (14.4) | <0.001 | 64 (6.8) | 53 (15.7) | <0.001 |
| Farmer (%)* | 3453 (68.7) | 1992 (62.2) | <0.001 | 612 (64.8) | 202 (59.8) | 0.097 |
| Current smoking (%) | 1498 (29.8) | 1391 (43.5) | <0.001 | 169 (17.9) | 77 (22.8) | 0.051 |
| History of disease (%) | | | | | | |
| Hypertension | 3098 (61.7) | 2032 (63.5) | 0.095 | 633 (67.1) | 213 (63.0) | 0.179 |
| Diabetes | 548 (10.9) | 342 (10.7) | 0.753 | 159 (16.8) | 54 (16.0) | 0.713 |
| Myocardial infarction | 328 (6.5) | 152 (4.8) | <0.001 | 66 (7.0) | 24 (7.1) | 0.946 |
| Angina | 600 (11.9) | 295 (9.2) | 0.102 | 107 (11.3) | 32 (9.5) | 0.343 |
| Heart Failure | 139 (2.8) | 31 (1.0) | <0.001 | 69 (7.3) | 10 (3.0) | 0.004 |
| Stroke | 461 (9.2) | 222 (6.9) | <0.001 | 121 (12.8) | 30 (8.9) | 0.054 |
| SBP <90 mmHg (%) | 145 (2.9) | 154 (4.8) | <0.001 | 79 (8.4) | 68 (20.1) | <0.001 |
| Heart rate ≥100 beats/min (%) | 644 (12.8) | 229 (7.2) | <0.001 | 261 (27.7) | 49 (14.5) | <0.001 |
| Continuous ECG monitoring (%) | 4388 (87.3) | 3099 (96.8) | <0.001 | 846 (89.6) | 329 (97.3) | <0.001 |
| In-hospital medication taken (%) | | | | | | |
| Aspirin | 4874 (97.0) | 3182 (99.4) | <0.001 | 882 (93.4) | 329 (97.3) | 0.007 |
| Clopidogrel | 4334 (86.3) | 3021 (94.4) | <0.001 | 737 (78.1) | 310 (91.7) | <0.001 |
| ACEI/ARB | 2948 (58.7) | 1940 (60.6) | 0.081 | 514 (54.5) | 169 (50.0) | 0.160 |
| β-Blockers | 3267 (65.0) | 2240 (70.0) | <0.001 | 473 (50.1) | 185 (54.7) | 0.144 |
| CCB | 399 (7.9) | 169 (5.3) | <0.001 | 110 (11.7) | 27 (8.0) | 0.061 |
| Statins | 4718 (93.9) | 3100 (96.8) | <0.001 | 830 (87.9) | 312 (92.3) | 0.027 |
| QCI intervention (%) | 2430 (48.4) | 1475 (46.1) | 0.044 | 452 (47.9) | 156 (46.2) | 0.585 |
| Time cycle (%) | | | | | | |
| 1 | 1060 (21.1) | 699 (21.8) | 0.110 | 212 (22.5) | 63 (18.6) | 0.629 |
| 2 | 946 (18.8) | 623 (19.5) | | 185 (19.6) | 69 (20.4) | |
| 3 | 1090 (21.7) | 614 (19.2) | | 178 (18.9) | 62 (18.3) | |
| 4 | 994 (19.8) | 655 (20.5) | | 199 (21.1) | 77 (22.8) | |
| 5 | 935 (18.6) | 610 (19.1) | | 170 (18.0) | 67 (19.8) | |

The results are presented as mean ± SD, median (quartile 1–quartile 3), or n (%).

*Baseline variable data missing: prehospital delay:1084 case, 11.4%; delay to fibrinolytic therapy, 231 cases, 6.5%; education: 1916 cases, 20.2%; occupation: 438 cases, 4.6%. The cases with missing data were assigned to an additional category.

[†]Calculated by using a t test, Wilcoxon rank test, or chi-square test.

eGFR, estimated glomerular filtration rate; SBP, systolic blood pressure; ECG, electrocardiograph; ACEI, angiotensin-converting enzyme inhibitors; ARB, angiotensin receptor blockers; CCB, calcium channel blockers; QCI, quality of care initiatives.

The results are presented as mean ± SD, median (quartile 1–quartile 3), or n (%).

Statistical analyses were conducted using SAS software, version 9.4 (SAS Institute Inc., Cary, NC, USA). All analyses were two-sided, with a *P*-value of 0.05 considered as the threshold for statistical significance.

## Results

### Baseline characteristics

The mean age of the 9508 patients with STEMI was 64.0±12.4 years, and 70.1% of participants were male. The median hospital stay was 10 days (interquartile range: 6 to 14 days). Of these STEMI patients, 1282 (13.5%) were classified as having CKD, and 3539 (37.2%) patients received fibrinolytic therapy. CKD Patients were less likely to receive fibrinolytic therapy than non-CKD patients (26.4% vs. 38.9%, *P*<0.001). Characteristics of STEMI patients who did and did not receive fibrinolytic therapy stratified by CKD are shown in Table 1.

### Associations between fibrinolytic therapy and outcomes

Crude incidence rates for outcomes by whether received fibrinolytic therapy are shown in S1 Table. Patients with CKD were more likely than those without RFK to experience a short-term MACE (19.7% vs. 5.6%), death (19.0% vs. 5.2%), recurrent MI (0.9% vs. 0.5%), stroke (0.4% vs. 0.1%), and severe bleeding (1.6% vs. 0.6%). The fully adjusted model in Table 2 shows that fibrinolytic therapy was significantly associated with a 13% reduction in the risk of short-term MACEs in patients without CKD but was not in those with CKD. The interaction term of fibrinolytic therapy with CKD complication was statistically significant (*P* = 0.026). We also found differential associations between fibrinolytic therapy and all-cause mortality in patients with and without CKD after multivariable adjustment (*P* for interaction = 0.012, S2 Table). The associations between fibrinolytic therapy and other secondary outcomes were not analyzed using regression analysis because of the small number of events.

### Association of fibrinolytic therapy with clinical outcomes stratified by the success/failure of fibrinolytic therapy

Among patients who received fibrinolytic therapy, fibrinolysis was classified as successful in 67.2% with and 83.1% without CKD, respectively. Our further analyses found that the success

**Table 2. Associations between fibrinolytic therapy and the risk of short-term major adverse cardiovascular events among patients with and without reduced kidney function (eGFR <60 mL/min/1.73 m$^2$).**

|  | eGFR ≥60 mL/min/1.73 m$^2$ (n = 8226) | | eGFR <60 mL/min/1.73 m$^2$ (n = 1282) | | *P* for interaction |
|---|---|---|---|---|---|
|  | **RR (95% CI)** | ***P* value** | **RR (95% CI)** | ***P* value** |  |
| Model 1* |  |  |  |  |  |
| No fibrinolysis | Ref | / | Ref | / | / |
| Fibrinolysis | 0.68 (0.52 to 0.89) | 0.006 | 1.03 (0.74 to 1.43) | 0.866 | 0.005 |
| Model 2† |  |  |  |  |  |
| No fibrinolysis | Ref | / | Ref | / |  |
| Fibrinolysis | 0.87 (0.76 to 0.99) | 0.034 | 1.02 (0.87 to 1.18) | 0.846 | 0.026 |

*Adjusted for age, sex, intervention, cycle, fibrin-specific thrombolytic agent, delay to admission, and delay to fibrinolytic therapy.

†Further adjusted for eGFR, education, occupation, current smoking, body mass index, history of diabetes mellitus, hypertension, cardiovascular disease, systolic blood pressure lower than 90 mmHg when presenting at hospital, heart rate higher than 100 beats/m when presenting at hospital, in-hospital use of aspirin, clopidogrel, angiotensin-converting enzyme inhibitors/angiotensin receptor blockers, β-Blockers, calcium channel blockers, and statins.

eGFR, estimated glomerular filtration rate; RR, relative risk; CI, confidence interval.

of fibrinolysis was associated with a lower risk of short-term MACEs, while the failure of fibrinolysis was associated with a higher risk, compared with patients not receiving fibrinolytic therapy (Table 3). After full adjustment (Fig 1), neither association was modified by renal function. The same results were found for all-cause mortality (see S3 Table).

### Prediction models for successful fibrinolysis

We further fit models on the success of fibrinolysis with its associated factors. The success of fibrinolysis was significantly associated with eGFR, thrombolytic agent, delay to fibrinolytic therapy, current smoking, high heart rate, and education in patients without CKD (Table 4). In patients with CKD, eGFR, thrombolytic agent, high heart rate, and education were still significantly associated with the success of fibrinolysis (Table 4).

### Sensitivity analysis

The propensity-score matching identified 588 patients with and 5502 patients without CKD. Baseline characteristics were comparable between patients treated with fibrinolysis and those who were not (S4 Table). The point estimators of the RR obtained using propensity score-matched subgroup were comparable with those from the main analyses, but the smaller sample size leads to wider 95% CIs. Thus, the association between fibrinolytic therapy and short-term MACEs and the modified effect of CKD were not significant anymore (S5 and S6 Tables). The associations of success/failure of fibrinolysis with short-term MACEs were still significant, and the modified effect of CKD on the associations remains non-significant (S7 and S8 Tables).

To limit the impact of AKI on our main results, we also performed analysis excluding 1565 participants with SBP <90 mmHg and/or heart rate ≥100 beats/minute, and this sensitivity analysis yielded similar results (S9 and S10 Tables).

### Discussion

The primary findings of this real-world study were that: (a) CKD modified the association between fibrinolytic therapy and short-term MACEs; (b) After fixing the status of the success of fibrinolytic therapy, CKD did not modify the above associations anymore. These findings strongly inform that future attentions should be paid to further optimize fibrinolytic therapy and increase the success rate among STEMI patients with CKD. On the one hand, guidelines for the management of patients with STEMI should recommend clinicians concern eGFR

**Table 3. Crude incidence rates of short-term outcomes by no, failed, and successful fibrinolysis, among patients with and without reduced kidney function (eGFR <60 mL/min/1.73 m$^2$).**

| | eGFR ≥60 mL/min/1.73 m$^2$ (n = 8226) | | | eGFR <60 mL/min/1.73 m$^2$ (n = 1282) | | |
|---|---|---|---|---|---|---|
| | No fibrinolysis (n = 5025) | Successful fibrinolysis (n = 2659) | Failed fibrinolysis (n = 542) | No fibrinolysis (n = 944) | Successful fibrinolysis (n = 227) | Failed fibrinolysis (n = 111) |
| MACEs (%) | 315 (6.3) | 69 (2.6) | 80 (14.8) | 177 (18.8) | 21 (9.3) | 54 (48.7) |
| All-cause mortality (%) | 295 (5.9) | 56 (2.1) | 77 (14.2) | 169 (17.9) | 20 (8.8) | 54 (48.7) |
| Recurrent MI (%) | 21 (0.4) | 15 (0.6) | 5 (0.9) | 7 (0.7) | 2 (0.9) | 3 (2.7) |
| Stroke (%) | 7 (0.1) | 2 (0.1) | 1 (0.2) | 5 (0.5) | 0 (0.0) | 0 (0.0) |
| Severe bleeding (%) | 22 (0.4) | 19 (0.7) | 5 (0.9) | 12 (1.3) | 4 (1.8) | 5 (4.5) |

The results are presented as n (%).

eGFR, estimated glomerular filtration rate; MACEs, major adverse cardiovascular events; MI, myocardial infarction.

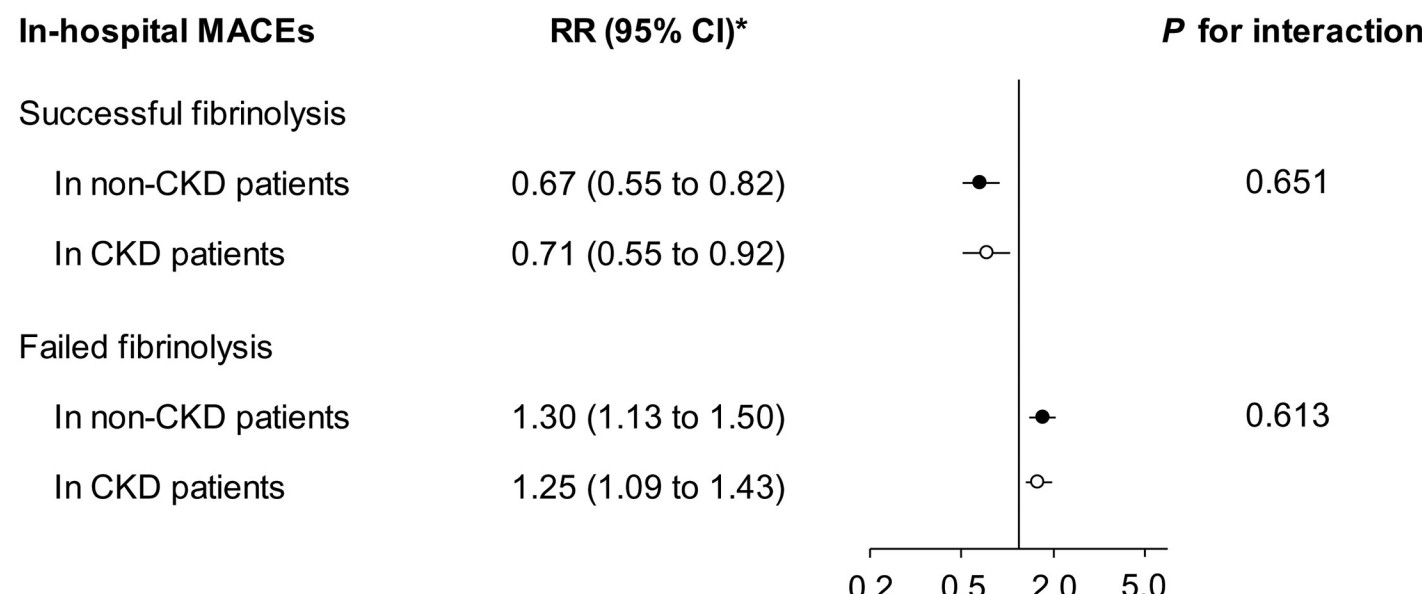

| In-hospital MACEs | RR (95% CI)* | P for interaction |
|---|---|---|
| **Successful fibrinolysis** | | |
| In non-CKD patients | 0.67 (0.55 to 0.82) | 0.651 |
| In CKD patients | 0.71 (0.55 to 0.92) | |
| **Failed fibrinolysis** | | |
| In non-CKD patients | 1.30 (1.13 to 1.50) | 0.613 |
| In CKD patients | 1.25 (1.09 to 1.43) | |

**Fig 1. Associations of successful and failed fibrinolysis with the risk of short-term major adverse cardiovascular events (MACEs) among patients with and without chronic kidney disease (CKD, eGFR <60 mL/min/1.73 m²), compared with those who did not receive fibrinolytic therapy.** *Adjusted for age, sex, intervention, cycle, fibrin-specific thrombolytic agent, delay to admission, delay to fibrinolytic therapy, eGFR, education, occupation, current smoking, body mass index, history of diabetes mellitus, hypertension, cardiovascular disease, systolic blood pressure lower than 90 mmHg when presenting at hospital, heart rate higher than 100 beats/m when presenting at hospital, in-hospital use of aspirin, clopidogrel, angiotensin-converting enzyme inhibitors/angiotensin receptor blockers, β-Blockers, calcium channel blockers, and statins. eGFR, estimated glomerular filtration rate; RR, relative risk; CI, confidence interval.

before fibrinolysis in clinical practice. To the best of our knowledge, this is one of the largest observational studies exploring the associations between fibrinolytic therapy and short-term outcomes among STEMI patients with and without CKD.

In Western countries, the incidence rate of STEMI is decreasing in recent decades. The age- and sex-adjusted incidence rates of STEMI decreased from 133 per 100 000 in 1999 to 50 per 100 000 in 2008 among the US population [3]. In Europe, an overall trend for a reduction in the incidence rate of STEMI has also been observed [2]. The incidence rate ranged from 43 to

**Table 4. Prediction models for successful fibrinolysis in patients who received fibrinolytic therapy among patients with and without reduced kidney function (eGFR <60 mL/min/1.73 m²).**

| Predictors | eGFR ≥60 mL/min/1.73 m² (n = 3201)* | | | eGFR <60 mL/min/1.73 m² (n = 338)*,† | | |
|---|---|---|---|---|---|---|
| | β | RR (95% CI) | P value | β | RR (95% CI) | P value |
| Intercept | −0.320 | / | <0.001 | −0.713 | / | <0.001 |
| eGFR, per 10 mL/min/1.73 m² in increase | 0.007 | 1.01 (1.00 to 1.01) | <0.001 | 0.059 | 1.06 (1.00 to 1.13) | 0.068 |
| Fibrin-specific thrombolytic agent (yes vs. no) | 0.103 | 1.11 (1.06 to 1.16) | <0.001 | 0.160 | 1.17 (1.02 to 1.34) | 0.022 |
| Delay to fibrinolytic therapy (yes vs. no) | −0.127 | 0.88 (0.83 to 0.94) | <0.001 | −0.261 | 0.77 (0.53 to 1.12) | 0.171 |
| Current smoking (yes vs. no) | 0.053 | 1.05 (1.02 to 1.09) | <0.001 | 0.048 | 1.05 (0.92 to 1.22) | 0.534 |
| Heart rate ≥100 beats/m (yes vs. no) | −0.146 | 0.86 (0.79 to 0.94) | 0.001 | −0.338 | 0.71 (0.54 to 0.95) | 0.019 |
| Education ≥High school (yes vs. no) | 0.068 | 1.07 (1.03 to 1.09) | 0.001 | 0.144 | 1.15 (1.00 to 1.34) | 0.054 |

*The models excluded age, sex, intervention, cycle, delay to admission, occupation, body mass index, history of hypertension, diabetes, cardiovascular disease, and systolic blood pressure lower than 90 mmHg when presenting at hospital, as these variables were not significant.

†The significant predictors in patients without reduced kidney function were still included in this model.

eGFR, estimated glomerular filtration rate; RR, relative risk; CI, confidence interval.

144 per 100 000 per year in European countries [2]. Conversely, the incidence rate of STEMI in China has substantially increased in recent years. The China PEACE-Retrospective Acute Myocardial Infarction Study reported that estimated national rates of hospital admission for STEMI per 100 000 people increased from 3·5 in 2001, to 7·9 in 2006, to 15·4 in 2011. If this keeps increasing, it is reasonable to expect that the incidence of STEMI in China will approach or even exceed that in Western countries. Therefore, assessment of fibrinolysis to the large population of STEMI patients in rural area of China and other developing countries in which STEMI patients cannot receive timely primary PCI is still significant and the impact of CKD on outcomes is relevant to clinical practice.

There is no specific recommendation of fibrinolytic therapy for patients with kidney disease in the latest guidelines for the management of STEMI [2, 3, 5], as few randomized controlled trials evaluated the treatment effect of the fibrinolysis in the subgroup of STEMI patients with CKD. This is one of the key reasons why STEMI patients presenting with CKD are less likely to receive fibrinolytic therapy [6, 10]. Consistent with our results, previous observational studies have reported that reperfusion rate was lower in STEMI patients with renal dysfunction [6], and highlighted the association between renal function and subsequent mortality rate in patients receiving fibrinolytic therapy [18–20]. However, whether CKD modifies the association between fibrinolytic therapy and consequent mortality was still unclear. In this large-scale study, we found that the strength of association between receipt of reperfusion therapy with fibrinolysis and short-term MACEs was significantly different in patients with and without CKD, which provides evidence and fills this gap to some extent. However, considering the nature limitations of observational study, randomized clinical trials performed in patients with CKD will be necessary to accurately evaluate the efficacy and safety of fibrinolytic therapy. For example, trials comparing the efficacy of PCI versus fibrin-specific thrombolytic agents on outcomes among patients with STEMI and CKD.

In current study, we found that successful fibrinolysis occurred less frequently in patients with CKD, likely-contributing to a higher risk of short-term MACEs. Our analyses further elucidated that CKD does not actually modify the relationships between successful and failed fibrinolysis with short-term MACEs. Therefore, the key question becomes how to increase the success rate of fibrinolytic therapy in STEMI patients with CKD. In order to answer this question, we built the prediction models of successful fibrinolysis and identified that eGFR, thrombolytic agent, high heart rate, and education were the crucial predictors in both groups. The success rate was only 67.2% in patients with CKD, while the rate was 83.1% in those without. When analysis was restricted to patients with CKD who had an eGFR $\geq$30 mL/min/1.73 m$^2$, received a fibrin-specific thrombolytic agent (tenecteplase or alteplase), had a heart rate <100 beats/m, and had been educated $\geq$high school, the success rate of fibrinolytic therapy was 87.5%.

Consistent with previous studies [6, 20], our study also found that patients with CKD were associated with a higher risk of bleeding events, which is another important challenging situation for clinicians to make choices regarding fibrinolytic therapy. Unfortunately, we cannot analyze the association between fibrinolytic therapy and severe bleeding events using regression analysis because of the small number of events. Further studies are needed to evaluate the safety of fibrinolytic therapy in STEMI patients with CKD.

Several underlying mechanisms may account for that CKD patients had a higher rate of failed fibrinolysis. Firstly, hypofibrinolysis has been well documented in patients with renal insufficiency and end-stage renal disease, which might lead to the higher rate of failed fibrinolysis [21]. A fibrin-specific thrombolytic agent should be the priority selection for CKD patients accepting fibrinolytic therapy. Therefore, clinicians should distinguish patients with known renal insufficiency at the time of fibrinolysis, from patients without renal insufficiency before deciding fibrinolysis. Secondly, CKD patients had more diffuse coronary artery disease

with a higher thrombotic burden than non-CKD patients [22]. Thirdly, CKD was associated with many of risk factors, such as diabetes, hypertension, and obesity, which might result in failed fibrinolysis [22].

Our study has several strengths. First of all, this is a large-scale, real-world study conducted in STEMI patients who received conservative treatments in non-PCI-capable hospitals. The primary and secondary endpoints were adjudicated by the independent committee, and the study process was closely monitored by a dedicated quality control team. In spite of these, our findings should be interpreted cautiously in the context of some potential limitations. First, the most important limitation is that this is an observational study of a treatment efficacy question. The possibility of residual confounding cannot be ruled out even after adjustment for a number of potential confounders. Although propensity score-matched subgroup analyses yielded similar results, unmeasured covariates might have influenced our results. Therefore, we are not able to infer any definite causal relationships due to the observational nature. Second, serum creatinine concentrations were measured at admission, thus we are not able to distinguish between CKD and AKI. Previous studies have shown that the underlining mechanisms between the two kidney diseases and consequent adverse events after MI might be different [7]. Although the sensitivity analysis excluding patients with high risk of AKI shows the similar results, the impact of AKI on our main results cannot be ruled out. Third, there is a possibility of selection bias, as 786 participants (7.6%) were excluded from the study because of missing data on serum creatinine concentration. As shown in S11 Table, the median hospital stay was significantly shorter and all-cause mortality was significantly higher in the 786 patients, which means that patients with absent creatinine were likely to have died before blood could be drawn. Therefore, there is a threat to the external validity of estimates and might limit the generalization of present findings to the original population. Fourth, the prediction models derived from this study have not been validated in an independent cohort. Finally, this study was conducted in non-PCI-capable hospitals in rural China, which highlights the need for caution in generalizing findings to other populations.

In conclusion, this real-world study suggests that CKD reduced the likelihood of successful fibrinolysis and increased the risk of short-term MACEs in patients with STEMI. Kidney function should be evaluated before performing fibrinolysis, and a fibrin-specific thrombolytic agent should be the priority selection for CKD patients when PCI is not available. Clinicians and researches should pay attention to developing new methods and techniques to improve the success rate of fibrinolytic therapy for STEMI patients with CKD. Randomized clinical trials performed in patients with STEMI and CKD will be necessary to accurately evaluate the efficacy and safety of intervention therapies.

## Supporting information

**S1 Table. Crude incidence rates of short-term outcomes by whether received fibrinolytic therapy among patients with and without chronic kidney disease.**
(DOCX)

**S2 Table. Associations between fibrinolytic therapy and all-cause death among patients with and without chronic kidney disease.**
(DOCX)

**S3 Table. Associations of successful and failed fibrinolysis with all-cause mortality among patients with and without chronic kidney disease.**
(DOCX)

**S4 Table. Baseline characteristics of propensity score-matched subgroup.**
(DOCX)

**S5 Table. Crude incidence rates of short-term outcomes by whether received fibrinolytic therapy among patients with and without chronic kidney disease, results of propensity score-matched subgroup.**
(DOCX)

**S6 Table. Associations between fibrinolytic therapy and the risk of short-term major adverse cardiovascular events among patients with and without chronic kidney disease, results of propensity score-matched subgroup.**
(DOCX)

**S7 Table. Crude incidence rates of short-term outcomes by no, failed, and successful fibrinolysis, among patients with and without chronic kidney disease, results of propensity score-matched subgroup.**
(DOCX)

**S8 Table. Associations of fibrinolytic therapy with risk of short-term major adverse cardiovascular events among patients with and without chronic kidney disease, results of propensity score-matched subgroup.**
(DOCX)

**S9 Table. Associations between fibrinolytic therapy and the risk of short-term major adverse cardiovascular events among patients with and without chronic kidney disease (eGFR $<$60 mL/min/1.73 m$^2$), results of sensitivity analysis excluding 1565 participants with SBP $<$90 mmHg and/or heart rate $\geq$100 beats/min.**
(DOCX)

**S10 Table. Associations of fibrinolytic therapy with risk of short-term major adverse cardiovascular events among patients with and without chronic kidney disease (eGFR $<$60 mL/min/1.73 m$^2$), sensitivity analysis excluding 1565 participants with SBP $<$90 mmHg and/or heart rate $\geq$100 beats/min.**
(DOCX)

**S11 Table. Comparison of baseline characteristics and outcomes between participants included (n = 9508) and excluded because of missing data on serum creatinine concentration (n = 768).**
(DOCX)

**S1 Checklist. STROBE statement—Checklist of items that should be included in reports of observational studies.**
(DOC)

# Acknowledgments

The CPACS-3 Study Steering Committee (sorted by the first letter of the family name):

Kalipso Chalkidou (National Institute for Health and Clinical Excellence), Runlin Gao* (Cardiovascular Institute and Fuwai Hospital), Dayi Hu (Peking University People's Hospital), Yong Huo (Peking University First Hospital), Yahui Jiao (NHFPC of China), Lingzhi Kong (NHFPC of China), Anushka Patel (The George Institute for Global Health), Eric Peterson (Duke Clinical Research Institute), Fiona Turnbull (The George Institute for Global Health),

Mark Woodward (The George Institute for Global Health), Yangfeng Wu* (The George Institute for Global Health at PUHSC).

*Co-principal investigators.

The independent endpoints adjudication committee (sorted by the first letter of the family name):

Yu Chen (Navy General Hospital of China), Chun Li (Peking University People's Hospital), Ming Shen (Peking University People's Hospital), Yihong Sun (Peking University People's Hospital), Yang Xi (Peking University People's Hospital).

## Author Contributions

**Data curation:** Wuxiang Xie, Yangfeng Wu.

**Formal analysis:** Wuxiang Xie, Lin Feng.

**Funding acquisition:** Runlin Gao, Yangfeng Wu.

**Investigation:** Wuxiang Xie, Yangfeng Wu.

**Methodology:** Wuxiang Xie, Anushka Patel, Eric Boersma, Runlin Gao, Yangfeng Wu.

**Project administration:** Yangfeng Wu.

**Resources:** Yangfeng Wu.

**Supervision:** Yangfeng Wu.

**Writing – original draft:** Wuxiang Xie.

**Writing – review & editing:** Wuxiang Xie, Anushka Patel, Eric Boersma, Lin Feng, Min Li, Runlin Gao.

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
