## [Decision Letter · Decision Letter 0]

30 Sep 2020

PONE-D-20-21065

Reduced kidney function and the outcomes of fibrinolysis for ST-segment elevation myocardial infarction: A real-world study

PLOS ONE

Dear Dr.  Wu

Thank you for submitting your manuscript to PLOS ONE. After careful consideration, we feel that it has merit but does not fully meet PLOS ONE’s publication criteria as it currently stands. Therefore, we invite you to submit a revised version of the manuscript that addresses the points raised during the review process.

The reviewers have expressed conflicting opinions about the priority of your paper. I found the paper of potential interest so I think It could be reasonable to give you the chance of improving the quality of your manuscript according to the issues raised by the reviewers.

Please submit your revised manuscript by  60 days. If you will need more time than this to complete your revisions, please reply to this message or contact the journal office at plosone@plos.org. Please include the following items when submitting your revised manuscript:

We look forward to receiving your revised manuscript.

Kind regards,

Simone Savastano

Academic Editor

PLOS ONE

Journal Requirements:

'Source of funding used to support the research and creation of the article is from Sanofi, China, through an unrestricted research grant. The George Institute for Global Health at PUHSC sponsored the study and owns the data. However, the authors are solely responsible for the design and conduct of this study, all study analyses, the drafting and editing of the manuscript, and its final contents. The funding source had no role in the design and conduct of the study; collection, management, analysis, and interpretation of the data; preparation, review, or approval of the manuscript; and decision to submit the manuscript for publication.'

We note that you received funding from a commercial source: Sanofi

Additional Editor Comments:

The reviewers have expressed conflicting opinions about the priority of your paper. I found the paper of some interest so I think It could be reasonable to give you the chance of improving the quality of your manuscript according to the issues raised by the Reviewers in particular of Reviewer #2

Reviewers' comments:

Reviewer's Responses to Questions

**Comments to the Author**

1. Is the manuscript technically sound, and do the data support the conclusions?

Reviewer #1: Yes

Reviewer #2: No

2. Has the statistical analysis been performed appropriately and rigorously? 

Reviewer #1: Yes

Reviewer #2: I Don't Know

3. Have the authors made all data underlying the findings in their manuscript fully available?

Reviewer #1: Yes

Reviewer #2: Yes

4. Is the manuscript presented in an intelligible fashion and written in standard English?

Reviewer #1: Yes

Reviewer #2: Yes

5. Review Comments to the Author

Reviewer #1: This is an interesting observational analysis from an established network, the Clinical Pathways in Acute Coronary Syndromes program (CPACS-3), a large study of the management of suspected acute coronary syndromes (ACS) in 101 county hospitals without onsite PCI capabilities in China.

The study is quite well presented, analyzed, and interpreted. Fibrinolysis remains an important reperfusion modality for many patients worldwide and these findings may have relevant implications for their management.

I have some suggestions for the author to consider:

MAJOR

For this research question it may be relevant to specify when the diagnosis of renal insufficiency was performed in relationship to the decision to perform fibrinolysis. Specifically, distinguish patients with known renal insufficiency at the time of fibrinolysis (such as patients on hemodialysis), patients with a new diagnosis of renal insufficiency when creatine was collected (immediately after the fibrinolysis I presume), from patients without renal insufficiency. This may help better understand the results and if, for patients with unknown renal function status, this information should be waited for before deciding fibrinolysis.

The discussion is well structured in general but might be improved by:

o Shortly speculating as to why an increased failure of fibrinolysis is observed in patients with RKF. Hypofibrinolysis for example has been well documented in patients with renal insufficiency and end-stage renal disease and could be mentioned, and the potential implications for different agents/dosage in this setting.

o What are the authors suggestions for managing these patients in terms of diagnosis (for eg. should we routinely wait creatinine results before performing fibrinolyis ?) and treatments (eg. considerations to accept a delay higher than 120 minutes to urgent angiography and primary PCI in these patients ?) .

MINOR

Reduced kidney function, RKF, is not a standard acronym. The authors may consider using chronic kidney disease or KDIGO equivalents.

www.clinicaltrails.gov under Trial registration is misspelled

Page 11, line 189: “The secondary outcome was all-cause mortality, recurrent MI, stroke, and severe bleeding”. The logic here, as for every composite and correctly reported for the primary endpoint is OR, that is “The secondary outcome was all-cause mortality, recurrent MI, stroke, OR severe bleeding”.

Reviewer #2: In this study the authors analyzed the data from a multicentric registry that involved a considerable number of acute coronary syndrome patients admitted to 101 country hospital in China without the capability to perform primary PCI.

The aim of the study was to assess whether reduced kidney function (RKF) may modify the association between fibrinolytic therapy and in-hospital major adverse cardiovascular events (MACE) – the primary endpoint.

The key results are the following:

1) patients with RKF were more likely to experience in-hospital MACEs compared to those with normal kidney function (19.7% vs 5.6%, respectively)

2) fibrinolytic therapy was associated with a 13% reduction in the primary endpoint in patients with normal kidney function (RR 0.87, 95% CI 0.76-0.99), but not in the those with RKF (RR 1.02, 95% CI 0.87-1.18)

3) Failed fibrinolytic therapy was higher in RFK patients compared to those with normal kidney function (32.8% vs 16.9%)

4) Failed fibrinolysis was not independently associated with the end-point at multivariate analysis (page 14, lines 259-261).

In the discussion (page 17, lines 324-325), the authors comment these results as follows: “in the current study, we found that successful fibrinolysis occurred less frequently in patients with RKF, likely contributing to a higher risk of in-hospital MACE”. The same thought is replicated in the conclusion of the abstract (“RFK modified the association of fibrinolytic therapy with the risk of in-hospital mortality via its negative association with successful fibrinolysis”, page 4, lines 64-65). However, this perspective seems to be contradicted by the results of multivariate analysis showing no independent association between adverse events and failed fibrinolysis and by the fact that in the subgroup with RKF fibrinolytic therapy was not associated with lower incidence of MACE. A possible explanation is that fibrinolytic therapy improves the prognosis in patients with normal kidney function but not in the RKF population (see point number 2) because the latter group is fragile and at-risk of adverse events independently from successful coronary reperfusion.

In my opinion, the most important finding of the study is that patients with RKF have a lower rate of successful fibrinolytic therapy and the authors should speculate in the discussion what led to this significant result (more diffuse coronary artery disease with a higher thrombotic burden in RKF patients?) and to its clinical implications (preference for ptca?).

Lastly, I am in disagreement with the inclusion in the primary endpoint (“in-hospital MACEs”) of patients who died within 1 week after a discharge against medical advice, under the category of all-cause mortality. Apart from being a contradiction itself, it introduces a significant bias since this entity cannot be measured.

6. PLOS authors have the option to publish the peer review history of their article (what does this mean?). If published, this will include your full peer review and any attached files.

Reviewer #1: No

Reviewer #2: No

---

## [Author Response · Author response to Decision Letter 0]

10 Nov 2020

Re：Reduced kidney function and the outcomes of fibrinolysis for ST-segment elevation myocardial infarction: A real-world study

(PONE-D-20-21065)

PLOS ONE

Response to Reviewer #1

Comment 1:

This is an interesting observational analysis from an established network, the Clinical Pathways in Acute Coronary Syndromes program (CPACS-3), a large study of the management of suspected acute coronary syndromes (ACS) in 101 county hospitals without onsite PCI capabilities in China.

The study is quite well presented, analyzed, and interpreted. Fibrinolysis remains an important reperfusion modality for many patients worldwide and these findings may have relevant implications for their management.

Response:

Thanks for your comments.

Change in the manuscript:

No change.

Comment 2:

I have some suggestions for the author to consider:

MAJOR

For this research question it may be relevant to specify when the diagnosis of renal insufficiency was performed in relationship to the decision to perform fibrinolysis. Specifically, distinguish patients with known renal insufficiency at the time of fibrinolysis (such as patients on hemodialysis), patients with a new diagnosis of renal insufficiency when creatinine was collected (immediately after the fibrinolysis I presume), from patients without renal insufficiency. This may help better understand the results and if, for patients with unknown renal function status, this information should be waited for before deciding fibrinolysis.

Response:

Accepted. In this study, creatinine was measured at the admission. Besides, we added a paragraph to further discuss our results according to your suggestions.

Change in the manuscript:

1. We clearly indicated creatinine was measured at the admission (Page 3, Lines 16 to 17, and Page 9, Line 19).

2. A paragraph has been added in the revised manuscript (Page 18, Lines 4 to 7): “A fibrin-specific thrombolytic agent should be the priority selection for CKD patients accepting fibrinolytic therapy. Therefore, clinicians should distinguish patients with known renal insufficiency at the time of fibrinolysis, from patients without renal insufficiency before deciding fibrinolysis”.

Comment 3:

The discussion is well structured in general but might be improved by:

Shortly speculating as to why an increased failure of fibrinolysis is observed in patients with RKF. Hypofibrinolysis for example has been well documented in patients with renal insufficiency and end-stage renal disease and could be mentioned, and the potential implications for different agents/dosage in this setting.

Response:

Thank you so much for this valuable comment. We added several sentences in the revised manuscript to describe the underlying mechanisms of higher failure rate of fibrinolysis in patient with CKD.

Change in the manuscript:

Sentences have been added in the revised manuscript to describe the underlying mechanisms of higher failure rate of fibrinolysis in patient with CKD (Page 18, Lines 1 to 10): “Several underlying mechanisms may account for that CKD patients had a higher rate of failed fibrinolysis. Firstly, hypofibrinolysis has been well documented in patients with renal insufficiency and end-stage renal disease, which might lead to the higher rate of failed fibrinolysis. A fibrin-specific thrombolytic agent should be the priority selection for CKD patients accepting fibrinolytic therapy. Therefore, clinicians should distinguish patients with known renal insufficiency at the time of fibrinolysis, from patients without renal insufficiency before deciding fibrinolysis. Secondly, CKD patients had more diffuse coronary artery disease with a higher thrombotic burden than non-CKD patients. Thirdly, patients with CKD could have more risk factors, such as diabetes, hypertension, and obesity, which might result in failed fibrinolysis”.

Comment 4:

What are the authors suggestions for managing these patients in terms of diagnosis (for eg. should we routinely wait creatinine results before performing fibrinolyis ?) and treatments (eg. considerations to accept a delay higher than 120 minutes to urgent angiography and primary PCI in these patients?) 

Response:

Thanks again. According to your suggestion, we added a paragraph to indicate our suggestions for managing patients with CKD in terms of diagnosis and treatments after STEMI.

Change in the manuscript:

We added a sentence to indicate our suggestions for managing patients with CKD in terms of diagnosis and treatments after STEMI (Page 19, Lines 16 to 18): “Kidney function should be evaluated before performing fibrinolysis, and a fibrin-specific thrombolytic agent should be the priority selection for CKD patients when PCI is not available”.

Comment 5:

MINOR

Reduced kidney function, RKF, is not a standard acronym. The authors may consider using chronic kidney disease or KDIGO equivalents.

Response:

Thank you for your suggestion. The term “reduced kidney function, RKF” has been modified to “chronic kidney disease, CKD” throughout this manuscript. 

Change in the manuscript:

The term “reduced kidney function, RKF” has been modified to “chronic kidney disease, CKD” throughout this manuscript. 

Comment 6:

www.clinicaltrails.gov under Trial registration is misspelled. 

Response:

We are sorry for this mistake. 

Change in the manuscript:

Done (Page 4, Line 16 and Page 9, Line 6).

Comment 7:

Page 11, line 189: “The secondary outcome was all-cause mortality, recurrent MI, stroke, and severe bleeding”. The logic here, as for every composite and correctly reported for the primary endpoint is OR, that is “The secondary outcome was all-cause mortality, recurrent MI, stroke, OR severe bleeding”. 

Response:

Accepted. 

Change in the manuscript:

The change has been made (Page 3, Lines 17 to 18 and Page 10, Line 22 to Page 11, Line 1). 

Response to Reviewer #2

Comment 1:

In this study the authors analyzed the data from a multicentric registry that involved a considerable number of acute coronary syndrome patients admitted to 101 country hospital in China without the capability to perform primary PCI.

The aim of the study was to assess whether reduced kidney function (RKF) may modify the association between fibrinolytic therapy and in-hospital major adverse cardiovascular events (MACE) – the primary endpoint.

The key results are the following:

1) patients with RKF were more likely to experience in-hospital MACEs compared to those with normal kidney function (19.7% vs 5.6%, respectively)

2) fibrinolytic therapy was associated with a 13% reduction in the primary endpoint in patients with normal kidney function (RR 0.87, 95% CI 0.76-0.99), but not in the those with RKF (RR 1.02, 95% CI 0.87-1.18)

3) Failed fibrinolytic therapy was higher in RFK patients compared to those with normal kidney function (32.8% vs 16.9%)

4) Failed fibrinolysis was not independently associated with the end-point at multivariate analysis (page 14, lines 259-261).

In the discussion (page 17, lines 324-325), the authors comment these results as follows: “in the current study, we found that successful fibrinolysis occurred less frequently in patients with RKF, likely contributing to a higher risk of in-hospital MACE”. The same thought is replicated in the conclusion of the abstract (“RFK modified the association of fibrinolytic therapy with the risk of in-hospital mortality via its negative association with successful fibrinolysis”, page 4, lines 64-65). However, this perspective seems to be contradicted by the results of multivariate analysis showing no independent association between adverse events and failed fibrinolysis and by the fact that in the subgroup with RKF fibrinolytic therapy was not associated with lower incidence of MACE. A possible explanation is that fibrinolytic therapy improves the prognosis in patients with normal kidney function but not in the RKF population (see point number 2) because the latter group is fragile and at-risk of adverse events independently from successful coronary reperfusion.

Response:

Thanks for your comments. We consider this perspective is not contradicted by the results that fibrinolytic therapy was not associated with lower incidence of MACE in patients with CKD. Actually, we found that failed fibrinolysis was associated with a similarly higher risk of in-hospital MACEs in both CKD and non-CKD patients. In short, CKD patients had a higher rate of failed fibrinolysis than non-CKD patients after STEMI, leading to CKD patients had a higher rate of in-hospital MACEs. Besides, stratified analyses showed that successful fibrinolysis was associated with a similarly lower risk among patients with and without CKD, while failed fibrinolysis was associated with a similarly higher risk in both CKD and non-CKD patients. Therefore, we consider that the higher rate of failed fibrinolysis in CKD patients is the key reason why CKD modified the association between fibrinolysis and in-hospital MACEs.

Change in the manuscript:

No change.

Comment 2:

In my opinion, the most important finding of the study is that patients with RKF have a lower rate of successful fibrinolytic therapy and the authors should speculate in the discussion what led to this significant result (more diffuse coronary artery disease with a higher thrombotic burden in RKF patients?) and to its clinical implications (preference for ptca?).

Response:

Accepted. Yes, the key point of this study is that patients with CKD had a lower success rate of fibrinolysis, the later in turn led to a significant higher risk of in-hospital MACEs. We add a paragraph to discuss the underlying mechanisms of higher failure rate of fibrinolysis in patient with CKD and its clinical implications.

Change in the manuscript:

The following sentences have been added in the revised manuscript (Page 18, Lines 1 to 10): “Several underlying mechanisms may account for the higher rate of failed fibrinolysis. Firstly, hypofibrinolysis has been well documented in patients with renal insufficiency and end-stage renal disease, which might lead to the higher rate of failed fibrinolysis. A fibrin-specific thrombolytic agent should be the priority selection for CKD patients accepting fibrinolytic therapy. Therefore, clinicians should distinguish patients with known renal insufficiency at the time of fibrinolysis, from patients without renal insufficiency before deciding fibrinolysis. Secondly, CKD patients had more diffuse coronary artery disease with a higher thrombotic burden than non-CKD patients. Thirdly, CKD was associated with many of risk factors, such as diabetes, hypertension, and obesity, which might result in failed fibrinolysis”.

Comment 4:

Lastly, I am in disagreement with the inclusion in the primary endpoint (“in-hospital MACEs”) of patients who died within 1 week after a discharge against medical advice, under the category of all-cause mortality. Apart from being a contradiction itself, it introduces a significant bias since this entity cannot be measured. Response:

Thanks for this comment. Yes, we agree with you that including those patients who died within 1 week after a discharge against medical advice might lead to bias of measuring outcome. However, if we did not include these out-of-hospital deaths, the bias would be greater, as Chinese people would prefer to die at home after relinquishing curative treatments.

Change in the manuscript:

No change.

---

## [Decision Letter · Decision Letter 1]

21 Dec 2020

PONE-D-20-21065R1

Chronic kidney disease  and the outcomes of fibrinolysis for ST-segment elevation myocardial infarction: A real-world study

PLOS ONE

Dear Dr. Yangfeng Wu

Thank you for submitting your manuscript to PLOS ONE. After careful consideration, we feel that it has merit but does not fully meet PLOS ONE’s publication criteria as it currently stands. Therefore, we invite you to submit a revised version of the manuscript that addresses the points raised during the review process.

ACADEMIC EDITOR:  Thank you very much for having revised your paper  whose quality has improved and for having addressed the majority of the Reviewers' comments.  However,  a statistical issue remains to ben solved as highlighted by Reviewer #2. I think that your paper could be considered for publication only if  you will be able to fix this problem and to address the minor comments of Reviewer #1

We look forward to receiving your revised manuscript.

Kind regards,

Simone Savastano

Academic Editor

PLOS ONE

Additional Editor Comments (if provided):

Thank you very much for having revised your paper whose quality has improved and for having addressed the majority of the Reviewers' comments. However, a statistical issue remains to ben solved as highlighted by Reviewer #2. I think that your paper could be considered for publication only if you will be able to fix this problem and to address the minor comments of Reviewer #1

Reviewers' comments:

Reviewer's Responses to Questions

**Comments to the Author**

1. If the authors have adequately addressed your comments raised in a previous round of review and you feel that this manuscript is now acceptable for publication, you may indicate that here to bypass the “Comments to the Author” section, enter your conflict of interest statement in the “Confidential to Editor” section, and submit your "Accept" recommendation.

Reviewer #1: All comments have been addressed

Reviewer #2: (No Response)

2. Is the manuscript technically sound, and do the data support the conclusions?

Reviewer #1: Yes

Reviewer #2: Partly

3. Has the statistical analysis been performed appropriately and rigorously? 

Reviewer #1: (No Response)

Reviewer #2: I Don't Know

4. Have the authors made all data underlying the findings in their manuscript fully available?

Reviewer #1: (No Response)

Reviewer #2: No

5. Is the manuscript presented in an intelligible fashion and written in standard English?

Reviewer #1: (No Response)

Reviewer #2: Yes

6. Review Comments to the Author

Reviewer #1: The authors have mostly and completely included my comments.

The wording of some sentences, such as the conclusions, may be improved. CKD is indeed and chronic and pre-existing conditions, not a complication. Why don't simply say something such as: "CKD reduced the likelihood of successful fibrinolysis and incresaed the risk of in-hospital MACEs in patients with STEMI (...)".

Reviewer #2: Dear authors, thank-you for your reply. However, there is still one major point that in my opinion needs classification: in the Table 1, a multi variable analysis showed that failed fibrinolysis is not an independent predictor of mace in patients with CKD. In figure 1, another multi variable analysis including the same co-variates gave completely opposite results: this time it seems that CKD has no role in modifying the RR, that it only depends on successful fibrinolysis. I admit I am not a statistician, but I think you should help me and other readers to understand the reason of this.

Another minor comment: if you included mortality post-voluntary discharge because of a cultural habit (Chinese people often desire so), you should state it in the methods section and maybe modify “in-hospital MACE” with “short-term MACE”

7. PLOS authors have the option to publish the peer review history of their article (what does this mean?). If published, this will include your full peer review and any attached files.

Reviewer #1: No

Reviewer #2: No

---

## [Author Response · Author response to Decision Letter 1]

4 Jan 2021

Re：Chronic kidney disease and the outcomes of fibrinolysis for ST-segment elevation myocardial infarction: A real-world study

(PONE-D-20-21065R1)

PLOS ONE

Response to Reviewer #1

Comment 1:

The authors have mostly and completely included my comments.

The wording of some sentences, such as the conclusions, may be improved. CKD is indeed and chronic and pre-existing conditions, not a complication. Why don't simply say something such as: "CKD reduced the likelihood of successful fibrinolysis and increased the risk of in-hospital MACEs in patients with STEMI (...)".

Response:

Thanks for your comments. We revised the conclusion and relevant sentences according to your suggestion (Page 4, Lines 11 to 12; Page 19, Lines 14 to 15).

Response to Reviewer #2

Comment 1:

Dear authors, thank-you for your reply. However, there is still one major point that in my opinion needs classification: in the Table 1, a multi variable analysis showed that failed fibrinolysis is not an independent predictor of mace in patients with CKD. In figure 1, another multi variable analysis including the same co-variates gave completely opposite results: this time it seems that CKD has no role in modifying the RR, that it only depends on successful fibrinolysis. I admit I am not a statistician, but I think you should help me and other readers to understand the reason of this.

Response:

Thanks for your comments. We have carefully checked the results in Table 1 and Figure 1. We found that Table 1 described the baseline characteristics of participants, so we suppose you actually saying is Table 2. 

In Table 2, our results showed that “fibrinolysis” (not failed fibrinolysis) is an independent predictor of MACEs in patients without CKD (P=0.034), but not in those with CKD (P=0.846). The interaction test was significant (P=0.026).

In Figure 1, fibrinolysis was classified as successful and failed fibrinolysis. Our further analyses found that CKD did not modify the relationships of successful and failed fibrinolysis with MACEs. 

To help the readers to better understand our article, we revised our main conclusion to “CKD reduced the likelihood of successful fibrinolysis and increased the risk of short-term MACEs in patients with STEMI” (Page 4, Lines 11 to 12; Page 19, Lines 14 to 15).

Comment 2:

Another minor comment: if you included mortality post-voluntary discharge because of a cultural habit (Chinese people often desire so), you should state it in the methods section and maybe modify “in-hospital MACE” with “short-term MACE”

Response:

Accepted. We added a sentence in the Methods section to state the reason why we included mortality post-voluntary discharge (Page 10, Lines 20 to 21). Besides, “in-hospital MACEs” has been revised to “short-term MACEs” throughout this manuscript.

---

## [Editor Report · Decision Letter 2]

5 Jan 2021

Chronic kidney disease  and the outcomes of fibrinolysis for ST-segment elevation myocardial infarction: A real-world study

PONE-D-20-21065R2

Dear Dr. Yangfeng Wu

We’re pleased to inform you that your manuscript has been judged scientifically suitable for publication and will be formally accepted for publication once it meets all outstanding technical requirements.

Kind regards,

Simone Savastano

Academic Editor

PLOS ONE

Additional Editor Comments (optional):

Thank you very much for having solved the issues raised by the reviewers. Albeit fibrinolysis is not the principal treatment for acute myocardial infarction worldwide however, it is still wildly used in some countries. I think it could important for readers at lest of those countries.
---

## [Editor Report · Acceptance letter]

8 Jan 2021

PONE-D-20-21065R2 

Chronic kidney disease and the outcomes of fibrinolysis for ST-segment elevation myocardial infarction: A real-world study 

Dear Dr. Wu:

I'm pleased to inform you that your manuscript has been deemed suitable for publication in PLOS ONE. Congratulations! Your manuscript is now with our production department. 

Kind regards, 

on behalf of

Dr. Simone Savastano 

Academic Editor

PLOS ONE